# Why Neighborhoods Matter: Traversal Context and Provenance in Agentic GraphRAG

Riccardo Terrenzi[1,*], Maximilian von Zastrow[1] and Serkan Ayvaz[1]

[1]*Centre for Industrial Software, University of Southern Denmark, Alsion 2, 6400 Sønderborg, Denmark*

### Abstract

Retrieval-Augmented Generation can improve factuality by grounding answers in external evidence, but Agentic GraphRAG complicates what it means for citations to be faithful. In these systems, an agent explores a knowledge graph before producing an answer and a small set of citations. We frame citation faithfulness as a trajectory-level problem: final citations should not only support the answer, but also account for the graph traversal, structure, and visited-but-uncited entities that may influence it. Through controlled ablation experiments, we compare the effects of isolating, removing, and masking cited and uncited graph entities. Our results show that cited evidence is often necessary, as removing it substantially changes answers and reduces accuracy. However, citations are not sufficient, because accurate answers can also depend on uncited traversal context and surrounding graph structure. These findings suggest that citation evaluation in Agentic GraphRAG should move beyond source support toward provenance over the broader retrieval trajectory.

### Keywords

agentic graphRAG, citation faithfulness, retrieval-augmented generation, knowledge graphs, provenance

## 1. Introduction

Although Large Language Models (LLMs) are becoming increasingly capable, factuality and faithfulness remain a fundamental problem due to hallucinations [1]. Hallucination is likely rooted in how LLMs are trained, since models are essentially rewarded for providing an answer rather than admitting uncertainty [2]. One solution is to ground answers through Retrieval-Augmented Generation (RAG) [3], where external sources supplement the model's own knowledge. In RAG, sources are typically retrieved from vector databases via semantic similarity. Graph RAG [4] instead retrieves directly from knowledge graphs, following typed entity-relation-entity triples via explicit paths, making retrieved evidence more structured and auditable. Beyond factual grounding, RAG enables a second advantage: attribution. When answers are grounded in retrieved sources, those sources can be cited, making claims verifiable by the reader. This makes RAG useful for improving both accuracy and transparency.

The latest development is the use of AI agents [5]: the combination of reasoning models and the ability to execute actions through tools enables LLMs to achieve a degree of autonomy. The LLM acts as a controller, making decisions based on observations and invoking tools, for example following the ReAct framework [6]. Applied to retrieval, this creates Agentic RAG: rather than a single retrieval step, the agent autonomously and iteratively decides what to query, inspects results, and issues further queries until it can produce a final answer with cited evidence. The same idea applies to Agentic Graph RAG, where the agent can traverse the knowledge graph autonomously to reach the answer.

This last setting raises a question about faithfulness and citation management. Prior work [7, 8] usually asks whether cited sources truly supported the answer, or whether the model's reliance on those sources was genuine [9]. Agentic GraphRAG raises an additional question: whether the cited entities are even a complete account of what the agent used to answer. As it traverses the graph, the agent has access to far more than the entities it ultimately cites—neighboring nodes, relation patterns, and community structure are all part of its working context, and any of them may shape its reasoning.

*Submitted at IJCAI-ECAI 2026 Joint Workshop on GENAIK and NORA*

*Corresponding author.

✉ rite@mmmi.sdu.dk (R. Terrenzi); maxvz@mmmi.sdu.dk (M. v. Zastrow); seay@mmmi.sdu.dk (S. Ayvaz)

🆔 0009-0006-8310-1940 (R. Terrenzi); 0000-0001-9848-3714 (M. v. Zastrow); 0000-0003-2016-4443 (S. Ayvaz)

Yet only some entities appear in the final citations. If the traversal as a whole meaningfully shapes the answer, then the citation set understates the model's evidential basis. This means that an audit framed only around "which sources were used" misses part of the graph context that influenced the response.

We hypothesize that, in Agentic GraphRAG, final citations capture only part of the information the agent relies on, and that the visited entities together with the structural context have a measurable effect on the accuracy and robustness of generated answers; to test this, we study the impact of cited entities, visited-but-uncited entities and graph traversal across three settings of Agentic GraphRAG.

The paper proposes the following contributions:

- We frame citation faithfulness in Agentic GraphRAG as a trajectory-level problem, where graph traversal, structure, and visited-but-uncited entities may be relevant to the answer.
- We introduce a graph-ablation methodology to test whether cited evidence is necessary, sufficient, and complete as an explanation of answer generation.
- We show that cited entities are often necessary but not sufficient, since accurate answers can depend on broader graph context not reflected in the final citations.

The code is available at: https://github.com/Ricter22/agentic_GraphRAG_rationalization.

## 2. Experimental Design

To investigate how agentic Graph RAG systems utilise visited entities together with the structural context they expose in their answers, we conduct a study over an established multi-hop QA benchmark. We build a knowledge base from the answer set and evaluate six systems to establish a performance baseline, ranging from a plain LLM to RAG, Graph RAG, and three agentic Graph RAG configurations (cf. Figure 1).

We then run three graph ablation studies with two experiments each, applied to all three agentic settings. The ablations examine the effect of isolating only cited evidence, removing cited evidence compared to removing a random set of entities, and removing visited-but-uncited entities (cf. Figure 2). Together these conditions let us assess how and to what extent each agent setting relies on the knowledge graph structural context when producing its answers. The remainder of this section describes each component in detail.

### 2.1. Dataset & Knowledge Base

We built a 30-question benchmark from the 2WikiMultiHopQA [10] development set. We chose this benchmark because it provides multi-hop questions with supporting facts, evidence triples, and distractor paragraphs, making it well suited for studying how agents use cited entities and graph structural context. We selected only 30 development-set questions to keep the experiments manageable while allowing qualitative inspection, filtering for bridge or comparison questions with short entity answers, at least two evidence triples, at least two supporting facts, and enough context. The subset covers 12 local-path questions, where the answer follows a clear gold evidence chain; 12 distractor-path questions, where overlapping distractor paragraphs create plausible wrong routes; and 6 summary-vs-local comparison questions, where the answer requires comparing or combining information across multiple entities.

The knowledge base was built from the selected questions and their associated paragraphs. We merged gold and distractor paragraphs into a corpus of 275 unique paragraphs, then chunked them into 318 text units. The graph was initialized from the dataset's annotated evidence triples, producing gold entities and relationships. We then enriched the graph by extracting additional entities and relationships from distractor text units. Finally, we detected graph communities using Leiden algorithm [11], generated community summaries, and built embedding indices for text units, entities, and communities. The resulting graph contains 1,815 entities, 1,692 relationships, and 7 communities.

## 2.2. Baseline and Agent Settings

We evaluate six QA LLM-based systems that differ in how much external context and graph-guided reasoning they expose to the model. The LLM used in all settings is `Mistral-Small-4-119B-2603`. We selected this model because it offers a favorable cost–performance trade-off while remaining capable both as a standalone question-answering model and as a tool-using agent.

The LLM baseline answers directly from the model's parametric knowledge, without retrieval. This setting gives us an idea of how many of the questions are already answerable just using the model's parametric knowledge. RAG retrieves the top-k (with $k = 5$ for the experiments) text units by embedding similarity and asks the model to answer from those passages only. GraphRAG is a non-agentic graph baseline: it retrieves relevant entities and communities, expands nearby relationships and linked text units, then gives this graph-derived context to the model in a single prompt. Agentic GraphRAG instead lets the model actively investigate the graph through tools: *search entities*, *get entity details*, *get neighbors*, *read textunit*, *search communities*, *read community*, and *submit answer*, allowing iterative traversal before producing an answer with cited entities, relationships, text units, and communities. This means that, for example, the agent can visit check an entity out without retrieving its text unit, or expand one node by checking its neighbours without actually visiting them. The two constrained variants use the same agentic GraphRAG environment but add provenance controls: in *visited only* when the agent submits an answer and citations are checked to see if the entities mentioned have also been visited; if not, the citations are rejected and the agent is asked to rephrase the response, while *evidence first* requires the agent to submit and validate its cited evidence before it is allowed to submit the final answer. These versions have been introduced because they allow us to test the agent's behavior and performance under conditions in which it is forced to handle citations without having the freedom, for example, to perform post-rationalization of citations or to cite false entities. The gold answers provided in the benchmark are simple and straightforward (e.g. names, places or dates). Hence, the answer correctness is checked through semantic matching.

## 2.3. Graph Ablation Studies

In order to study the effect of graph structure on the citation and answering behavior of our agentic Graph RAG settings, we run six graph ablation experiments organized into three studies. All experiments follow a common procedure. Given the trace from the baseline run, we categorize every entity in the graph into three groups for each individual question: entities that were not visited, entities that were visited and cited, and entities that were visited but not cited (cf. Figure 2). We then construct a question-specific modified graph according to these labels by removing entities or restricting access to their linked text units, and run a fresh agent instance on the same question with the modified graph.

### Study 1 - Isolation of Evidence

To measure how much of the answer can be attributed to the cited evidence alone, we restrict each agent to its originally cited evidence in two ways. In **Full Isolation** every entity except those cited in the baseline is removed from the graph, leaving the agent with only the cited evidence to generate its answer. If the agent still answers correctly, the cited evidence was sufficient. If accuracy drops, the original answer depended on non-cited entities, text units, or relationships. In **Text-only Isolation** no entities are removed, but access to text units attached to non-cited entities is blocked. The metadata of the non cited entities in this case is available, allowing the agent to collect traversal information. The graph structure remains intact, and the agent can still traverse entity nodes and their neighbors, but can only read text units from cited entities. If accuracy rises back, compared to full isolation, it shows the impact of preserving the graph structure.

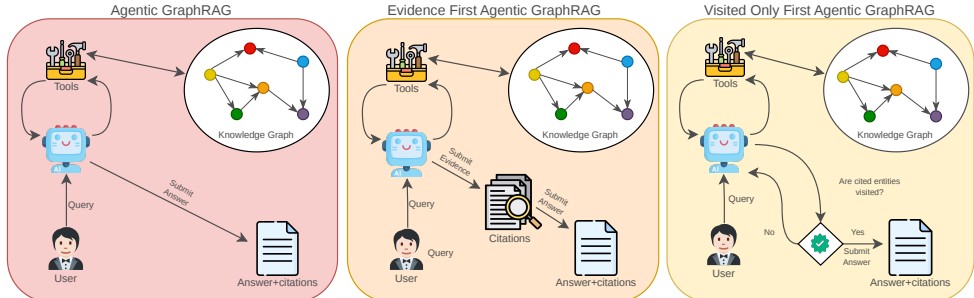

**Figure 1:** Representation of the three agentic graphRAG systems we test in our experimental design.

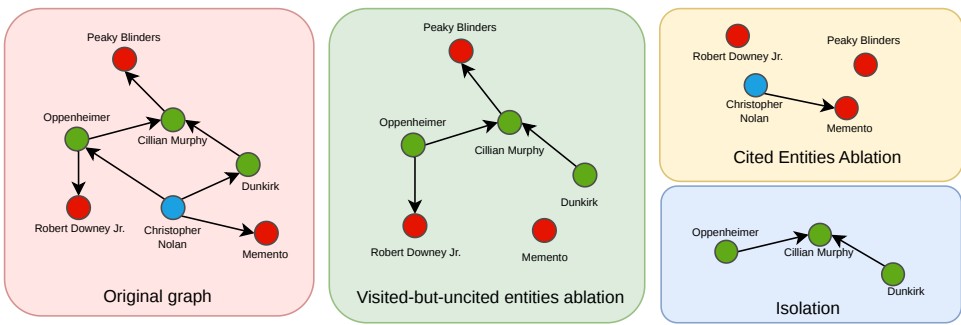

**Figure 2:** Example of the three graph ablations effect on a synthetic subgraph. Cited entities are represented in green, visited but not cited in blue, not visited in red.

## Study 2 - Cited Evidence Ablation

Where Study 1 tests whether cited evidence is sufficient, Study 2 tests whether it is necessary. We perform another two experiments. In **Cited Removal** every entity cited in the baseline is removed from the graph, this process is performed singularly and specifically for each single question. The agent must now answer without the evidence it explicitly attributed, while the rest of the graph structure remains intact. Simply observing that removing cited evidence degrades accuracy is not enough to conclude that citations are meaningful — any removal of entities from the graph reduces the information available to the agent and may degrade accuracy for structural reasons alone. In **Random Removal** we additionally removed a same-sized random set of non-cited but plausibly retrievable text units. Since the number of removed entities is the same, any difference in accuracy between the two conditions can be attributed specifically to the informational value of the cited entities rather than to the structural disruption caused by removing nodes from the graph.

## Study 3 - Visited-but-uncited entities Ablation

Study 3 examines the role of entities the agent visited but did not cite. During multi-step graph exploration, the agent follows edges and inspects neighbors to navigate toward an answer, yet many of the visited nodes never appear in the final citation. If removing these entities degrades accuracy, the most direct explanation is that the agent relied on evidence it did not cite. However, it may also indicate that the traversal path itself shaped the answer through the context it exposed during navigation or the structure it used to reach the cited evidence. To test the second theory, similarly to Study 1 we also compared the effect of fully removing the entities in **Entity Removal** or only restricting access to the text in **Entity text mask**. In the **Entity text mask**, differently compared to **Text-only Isolation**, the metadata of the blocked entities will not be available in order to avoid any information leakage from the traversal context.

**Table 1**

Baseline experiment accuracy and evidence-usage footprint. TUs stands for text units.

| System | Accuracy | Retrieved TUs | Cited TUs | Visited entities | Cited entities |
|---|---|---|---|---|---|
| **Agentic GraphRAG** | 76.0% | 1.5 | 1.6 | 11.9 | 1.9 |
| *evidence first* | **80.0**% | 1.4 | 1.3 | 10.5 | 1.8 |
| *visited only* | 72.0% | 1.6 | 1.3 | 11.1 | 1.6 |
| **GraphRAG** | 60.0% | 13.3 | 1.4 | 15.6 | 1.8 |
| **RAG** | 56.7% | 5.0 | 1.0 | - | - |
| **LLM** | 16.7% | - | - | - | - |

**Table 2**

Intervention results by system after excluding the five questions answered correctly by the single LLM baseline. "Output changed" reports the percentage of changed answers after intervention. Parentheses in "Output changed" rows report originally correct answers that stayed correct after intervention out of originally correct answers. "Acc." denotes post-intervention accuracy. Parentheses in "Acc." rows report correctly answered questions out of total. Arrows represent symbolically the accuracy delta for each setting between intervention and baseline. Fractional counts in "Random ablation" represent the average from 3 random runs.

| Condition | Outcome | Agentic GraphRAG | Evidence-first | Visited-only | Simple GraphRAG |
|---|---|---|---|---|---|
| *Baseline experiment* | | | | | |
| Baseline | Acc. | 76.0% (19/25) | 80.0% (20/25) | 72.0% (18/25) | 60.0% (15/25) |
| *Cited entities ablation* | | | | | |
| Cited ablation | Output changed | 76.0% (6/19) | 72.0% (7/20) | 68.0% (7/18) | 80.0% (5/15) |
| | Acc. | ↓ 36.0% (9/25) | ↓ 32.0% (8/25) | ↓ 40.0% (10/25) | ↓ 28.0% (7/25) |
| Random ablation | Output changed | 30.7% (18/19) | 30.7% (16.3/20) | 36.0% (15.7/18) | 53.3% (10.7/15) |
| | Acc. | ↑ 84.0% (21/25) | ↓ 76.0% (19/25) | ↑ 80.0% (20/25) | ↓ 58.7% (14.7/25) |
| *Isolation* | | | | | |
| Full isolation | Output changed | 40.0% (14/19) | 68.0% (7/20) | 80.0% (5/18) | 36.0% (12/15) |
| | Acc. | ↓ 68.0% (17/25) | ↓ 28.0% (7/25) | ↓ 24.0% (6/25) | ↓ 48.0% (12/25) |
| Text-only isolation | Output changed | 20.0% (18/19) | 36.0% (15/20) | 40.0% (15/18) | 24.0% (15/15) |
| | Acc. | ↓ 72.0% (18/25) | ↓ 60.0% (15/25) | ↓ 64.0% (16/25) | →60.0% (15/25) |
| *Visited-but-uncited entities ablation* | | | | | |
| Entity removal | Output changed | 32.0% (17/19) | 36.0% (15/20) | 44.0% (13/18) | 28.0% (14/15) |
| | Acc. | ↓ 68.0% (17/25) | ↓ 68.0% (17/25) | ↓ 60.0% (15/25) | ↓ 56.0% (14/25) |
| Entity text mask | Output changed | 40.0% (16/19) | 48.0% (12/20) | 52.0% (12/18) | 72.0% (6/15) |
| | Acc. | ↓ 72.0% (18/25) | ↓ 60.0% (15/25) | ↓ 52.0% (13/25) | ↓ 40.0% (10/25) |

# 3. Results & Discussion

The findings indicate that citation faithfulness in agentic GraphRAG is neither binary nor adequately characterized by checking whether final citations support the generated answer, since citations are behaviorally meaningful but incomplete: they often identify evidence whose removal changes the answer or reduces accuracy, yet omit broader graph context involved in the agent's reasoning.

Table 1 shows that agentic systems typically visit ten to twelve entities while citing only around two, creating a gap between the graph-interaction trace and the final provenance trace. This gap is not itself evidence of unfaithfulness, since a visited entity may have been surfaced by a graph tool without being explicitly used by the model, but it raises a question of provenance sufficiency: are final citations enough to explain answer behavior, or does uncited graph context remain behaviorally relevant? To reduce the influence of parametric knowledge, we exclude the five questions answered correctly by the LLM-only baseline from subsequent experiments, leaving 25 questions that more directly isolate the role of external graph context.

The *Cited Entities ablation* shows that final citations are not merely decorative: removing cited entities

produces frequent answer changes and a substantial accuracy drop across all systems, indicating that cited entities often contain evidence important for preserving the original answer. This effect is not simply due to graph perturbation, since the *Random Entities ablation* does not produce a comparable drop and sometimes improves performance by reducing graph noise. At the same time, systems still answer correctly in more than 30% of cases after cited entities are removed, suggesting that cited entities are important but not sufficient to fully account for answer generation.

The *Isolation* experiments further distinguish citation support from provenance sufficiency. When systems are restricted to only the originally cited entities, accuracy decreases across all settings, with especially large drops in the constrained agentic variants and more moderate but still present drops for *Agentic GraphRAG* and *GraphRAG*. This indicates that final citations alone do not robustly reconstruct the context needed for accurate answering, a gap we attribute to traversal context: visited-but-uncited entities, their relations, local neighborhoods, and structural cues.

The *Text-only Isolation* condition supports this interpretation. When graph structure is preserved but text units from non-cited entities are masked, both accuracy and answer stability improve relative to full isolation. This suggests that even without non-cited textual content, the presence, position, and connectivity of uncited entities can guide traversal, constrain the search space, or provide structural signals that influence answer generation.

Finally, the *visited-but-uncited ablation* interventions show that uncited context is not merely incidental: both entity removal and text masking reduce performance across all settings, although less severely than in the isolation experiments. This indicates that visited-but-uncited entities contribute through textual content, metadata, or structural context. In the *Entity Text Mask* condition, masking also removes entity metadata, likely explaining the additional decrease in accuracy and robustness observed in most settings.

Taken together, these results distinguish citation correctness from provenance sufficiency. A citation may support the final answer while omitting context that was behaviorally important during generation. Faithfulness and provenance for agentic graph-based systems therefore require accounting not only for final cited entities, but also for the broader graph-interaction trace: visited-but-uncited entities, neighborhood structure, traversal paths, and implicit structural signals that shape the agent's answer.

## 4. Conclusion

This paper examined whether final citations in Agentic GraphRAG are sufficient to explain the evidence basis of generated answers, and our findings suggest that they are not. While cited entities often play a necessary role, accurate answering can also depend on graph traversal, neighborhood structure, and visited-but-uncited entities that influence how the agent discovers, selects, and interprets evidence.

The results show that citation faithfulness in Agentic GraphRAG requires a broader notion of provenance. A citation set may correctly identify supporting evidence while omitting parts of the retrieval trajectory that contributed to the answer. Faithful citation mechanisms should therefore account not only for cited sources, but also for the graph context and retrieval path that shape the final response, treating provenance as a trajectory-level rather than final-output property.

## 5. Limitations

This study is limited by the small benchmark size and by its use of a controlled knowledge graph built from 2WikiMultiHopQA rather than a large real-world knowledge graph. Future work should evaluate the same interventions on larger and richer graph structures, and should develop citation mechanisms that expose not only final supporting entities but also the relevant traversal context and trajectories.

## Declaration on Generative AI

During the preparation of this work, the author(s) used Claude (Anthropic) and ChatGPT (OpenAI) in order to: drafting content, improve writing style, generate literature review, paraphrase and reword, and content enhancement. After using this tool/service, the author(s) reviewed and edited the content as needed and take(s) full responsibility for the publication's content.

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
