# OpenReview forum: "Why Neighborhoods Matter: Traversal Context and Provenance in Agentic GraphRAG"
_ijcai.org/IJCAI-ECAI/2026/Workshop/GENAIK-NORA — IJCAI-ECAI 2026 Joint Workshop on GENAIK and NORA_

### Official Review · Reviewer_967T · 2026-05-19
**Important research question with limitation experiments and theoretical justification**

**Rating:** 5
**Confidence:** 4

**Review:**

Summary:
This paper suggests the exploration of citation faithfulness by Agentic GraphRAGs. The paper studies the necessity and sufficiency of citation graphs. Knowledge graphs are traversed autonomously by Agentic GraphRAG systems and the entities, relationships, graph structures are used by the agent as context. While Agentic GraphRAG returns a list of citations, it does not provide the whole list of visited nodes.The paper proposes an ablation methodology to assess the necessity, sufficiency, and completeness of the returned citations. Three studies on isolation of evidence, citation removal, and visited by uncited entities ablation are described. Experiments are performed on GraphRAG, Agentic GraphRAG, and two variants of GraphRAG where returned citations need to be visited by the agent and the citations need to be submitted and validated before the response is accepted. A 30-question benchmark was created from the 2WikiMultiHopQA dataset.

While the ideas are worth pursuing, weak empirical and theoretical evidence reduce the impact of this paper. Revising this paper and going into depth on the analysis of the citation graphs could greatly improve it.

Pros:
The problem is well-motivated and the proposed graph-ablation methodologies could be very useful.
The paper is easy to understand and the concepts are explained well.
The figures work well to illustrate the paper.

Cons:
The ideas are interesting but theoretical analysis is missing. While the problem is intuitively explained, the proposed methods do not include theorems or proofs to justify why this design of the studies is the optimum one for each hypothesis.
Experimentation is limited to one dataset and 30 questions.
The results table is hard to read.

Details:
For example, Random Removal sounds plausible but can you provide theoretical proof that it eliminates the performance degradation due to structural disruption?
Similarly, while empirical evidence is provided, there is a lack of theoretical justification of the necessity vs sufficiency vs completeness of citations returned which makes it hard to generalize the problem.
I’m also curious about constructing the citation tree (assuming it’s a DAG) based on the visited nodes by the agent and analyzing the resulting graph. How much does it vary across questions? Is there a correlation between tree-depth and accuracy of the different variants?
Can you also provide more details on how the evidence-first and visited-only variants are implemented? What information is provided to the agent when the citations are rejected? Does it have to start answering the question from scratch?

---

### Official Review · Reviewer_RnG7 · 2026-05-24
**Why Neighborhoods Matter: Traversal Context and Provenance in Agentic GraphRAG**

**Rating:** 4
**Confidence:** 5

**Review:**

**Overview**

This paper investigates citation faithfulness in Agentic GraphRAG systems by framing it as a trajectory-level problem rather than a final-output property. Through controlled graph ablation experiments on a subset of the 2WikiMultiHopQA dataset, the authors demonstrate that while cited evidence is necessary, accurate answers also heavily depend on uncited traversal context and surrounding graph structure.

**Strengths**

1. The conceptual shift from evaluating citation faithfulness solely based on final outputs to considering the entire retrieval trajectory provides a more comprehensive framework for auditing Agentic RAG systems.

2. The experimental design using three distinct graph ablation studies (isolation, cited evidence ablation, and visited-but-uncited entities ablation) effectively isolates the impact of different graph structural components on model performance.

**Weaknesses**

1. The benchmark consists of only 30 questions from 2WikiMultiHopQA. This small sample size limits the statistical significance and generalizability of the findings. The authors should expand the evaluation to a larger and more diverse set of questions to validate the claims.

2. The experiments rely exclusively on a single language model (Mistral-Small-4-119B-2603). Different LLMs exhibit varying degrees of reliance on parametric knowledge versus retrieved context, which could skew the ablation results. Testing across multiple models is necessary to prove the phenomenon is model-agnostic.

3. Although 5 questions answered correctly by the LLM baseline were excluded, the remaining 25 questions might still be partially influenced by the model's internal knowledge. When partial graph context is provided, the model might use it to trigger latent parametric knowledge rather than relying solely on the graph. A more rigorous control, such as using a fully synthetic knowledge graph, is needed.

4. The paper lacks a precise formalization of what constitutes a "visited" entity during the agent's traversal. It is unclear if an entity is considered visited simply by appearing in a neighbor list or if the agent must explicitly query its text unit. Clarifying this definition is crucial for reproducibility and understanding the agent's behavior.

5. The performance of agentic systems is highly sensitive to the specific prompts and tool descriptions provided to the LLM. The paper does not detail the prompt structures used for the three agentic configurations. Providing these details in an appendix would strengthen the methodology and allow for proper replication.

6. The paper proposes that citation evaluation should move toward the broader retrieval trajectory but does not introduce a quantitative metric for this. Without a concrete metric, it is difficult to systematically compare different Agentic GraphRAG systems based on trajectory faithfulness. Proposing a calculable metric would significantly improve the paper's contribution.

7. The process of enriching the graph with distractor text units and detecting communities is briefly mentioned but lacks specific parameters. The structural properties of the resulting graph directly impact the ablation results. The authors should provide the exact hyperparameters and thresholds used during graph construction.

8. The paper reports accuracy drops and output changes but lacks a qualitative error analysis of the failed cases. Understanding whether the agent failed due to hallucination, premature stopping, or tool misuse after ablation is essential. Adding a breakdown of failure modes would provide deeper insights into how traversal context affects reasoning.

---

### Official Review · Reviewer_MNLr · 2026-05-30
**Interesting investigation of the role of graph structure in Agentic GraphRAG showing that the structure and non-cited nodes are key to improved accuracy.**

**Rating:** 8
**Confidence:** 4

**Review:**

This paper provides an interesting investigation of the role of graph structure in Agentic GraphRAG showing that the structure and non-cited nodes are key to improved accuracy.  Understanding the way graph structures and citations (attribution) affects RAG, especially agentic RAG, is very important and this study is an excellent step towards that.

A small number of non-trivial questions are selected from a baseline dataset (2WikiMultiHopQA)  and a small KG constructed for this, including distractor nodes/entities (1815 nodes; 1692 edges/relations; 7 communities).
The paper then compares 6 approaches. Two baselines are LLM only (a Mistral model used for all the approaches) and standard RAG. A graph RAG version and 3 agentic GraphRAGs (base, evidence first, visited only).
Once the base performance of each approach is determined, 3 studies are done.  Study 1: 2 forms of isolating evidence by keeping only the cited entities and by removing the text from the non-cited entities. Study 2: removing the cited entities; random removal. Study 3: different removal strategies for entities that the agent visited but did not cite.

Evidence-first agentic RAG outperforms the other models, with the LLM baseline performing very poorly and simple RAG better but not nearly as well as the graph approaches (probably due to the selection of difficult questions). The results indicate that the graph traversal of the agent is important, even if visited nodes are not cited.

Pros:
- Very timely area to investigate.  Well designed study.
- Good mix of baselines and versions of agentic RAG.
- Detailed study/ablations of which nodes (cited or just visited) were important to determining the answers.
- Paper is well written and easy to follow.

Cons/areas for improvement:
- Can the dataset be made available for reproducibility?  Ideally also the code used to run the experiment.
- Having an example of the type of question and answer and entities cited and visited would help in understanding the task, including exactly what types of questions were selected for the study.
- How was correctness judged?  That is, were the answers extremely simple (e.g. just a person name) and hence trivial to string match or did an LLM compare to the gold or did a person compare to the gold?

Other minor comments:
- Float the figures closer to where they are discussed.
- I believe you still have space but if you need more, fig 1 does not need to have the cute pictures and so could be compacted to just text and links.
- Citation and attribution are extremely important. This is discussed in the introduction, but the authors under-sell the impact of this.  Also, this paper provides general evidence on the usefulness of graph-based approaches, something which is cited elsewhere in the literature, but it is good to have newer evidence on agentic approaches.
- sec 2.2: for the RAG baseline, was there any similarity threshold used? how many documents were selected (e.g. was there a top-k used)?
- Table 2: in the row on Cited entities ablation/Random ablation/Output changed, what due the numbers like 16.3, 15.7, and 10.7 mean? how can there be a decimal version of an answer changed.  I suspect this is a typo from an earlier format.

---

### Official Review · Reviewer_zeVg · 2026-06-06
**Weak Accept, Lack of sufficient evidence**

**Rating:** 6
**Confidence:** 4

**Review:**

Good framing and a thoughtful control, but the evidence is thin and the work is diagnostic rather than constructive. Three fixes would lift it: 1.  define the grading metric + add citation precision/recall vs gold
2. scale to ≥2 models and a larger/real KG with seeds and CIs
3.  propose one trajectory-provenance measure and show it helps.

** Quality **
1. Strong idea, thin evidence. 30 questions → 25 after filtering, one model (Mistral-Small-4-119B-2603), one dataset (2WikiMultiHopQA [10]), apparently one run per condition. At n=25 each question is ~4 points, so the headline cited-removal drop ("↓36%", 9/25) rests on about nine questions. No confidence intervals or significance tests.
2. The Random ablation row shows fractional counts (e.g. 16.3/20), implying averaging over runs for that row only -- inconsistent with the rest of Table 2. Please state the protocol.
3. The grading metric is never defined -- exact match, F1, or an LLM judge? For a measurement paper this must be specified, ideally with code/prompts.
4. Best choice: the random-removal control, which separates real information loss from generic graph damage. Good instinct.
5. Most natural metric is missing -- you have gold supporting facts but never report citation precision/recall against them.
6. Table 1 looks off: cited TUs (1.6) > retrieved TUs (1.5) for Agentic GraphRAG. Define how a unit can be cited but not retrieved.

** Clarity **
1. Generally well written and easy to follow; the necessary/sufficient/complete framing and Figures 1-2 help.
2. Table 2 is hard to read -- the parenthetical semantics ("stayed correct out of originally correct") need a clearer caption.
3. No related-work section and only 11 references; the GenAI declaration notes the lit review was AI-generated, and it shows.
4. nit: the copyright still says "© 2022".

** Originality **
1. The framing -- citations as an incomplete account of the agent's evidential basis, i.e. trajectory-level provenance -- is the genuine contribution, and a useful one.
2. It sits close to Wallat et al. [9] ("correctness ≠ faithfulness"), and the method is standard leave-one-out / counterfactual ablation applied to graph entities. The evidence-first and visited-only variants are a nice touch.
3. The idea is named, not solved: no new metric, mechanism, or system is proposed.

** Significance **
1. Real, timely problem -- auditability of agentic RAG matters, and the citation-support vs provenance-sufficiency distinction is worth attention.
2. Impact is limited by scale/scope: a 1,815-entity graph built from 2WikiMultiHopQA is small and synthetic, not a real-world KG.
3. One result deserves more: evidence-first is the best baseline (80%) yet collapses most under full isolation (↓28%, Table 2). Interesting and unexplored.
4. It would matter much more with even one concrete trajectory-provenance metric shown to beat plain source-support.

---

### Decision · Program_Chairs · 2026-06-10

Accept